# Identification and Characterization of Three Novel Solemo-like Viruses in the White-Backed Planthopper, *Sogatella furcifera*

**DOI:** 10.3390/insects15060394

**Published:** 2024-05-28

**Authors:** Jing-Na Yuan, Zhuang-Xin Ye, Meng-Nan Chen, Peng-Peng Ren, Chao Ning, Zong-Tao Sun, Jian-Ping Chen, Chuan-Xi Zhang, Jun-Min Li, Qianzhuo Mao

**Affiliations:** 1State Key Laboratory for Managing Biotic and Chemical Threats to the Quality and Safety of Agro-Products, Key Laboratory of Biotechnology in Plant Protection of Ministry of Agriculture and Zhejiang Province, Institute of Plant Virology, Ningbo University, Ningbo 315211, China; yjn2430304122@163.com (J.-N.Y.); yzx244522794@163.com (Z.-X.Y.); 15066716480@163.com (M.-N.C.); 2011074040@nbu.edu.cn (P.-P.R.); n1ngchao@163.com (C.N.); sunzongtao@nbu.edu.cn (Z.-T.S.); jianpingchen@nbu.edu.cn (J.-P.C.); chxzhang@zju.edu.cn (C.-X.Z.); lijunmin@nbu.edu.cn (J.-M.L.); 2College of Forestry, Nanjing Forestry University, Nanjing 210037, China

**Keywords:** *Sogatella furcifera*, insect-specific virus, *Solemoviridae*, Sogatella furcifera solemo-like virus

## Abstract

**Simple Summary:**

White-backed planthoppers (WBPH; *Sogatella furcifera*, Hemiptera: Delphacidae) serve as significant vectors for rice viruses and harbor various insect-specific viruses (ISVs). In this study, we identified three previously unknown viruses that infect WBPH, which are classified under the plant-associated viral family *Solemoviridae*. Interestingly, these viruses were assigned to an unclassified clade, indicating the presence of novel viral taxa within the *Solemoviridae* family. To further explore the characteristics of these viruses, we analyzed vsiRNA (virus-derived small interfering RNA) patterns. The analysis revealed that Sogatella furcifera solemo-like virus 1 (SFSolV1) and Sogatella furcifera solemo-like virus 2 (SFSolV2) exhibit typical dicer-mediated patterns of vsiRNA, similar to those found in leafhoppers, grasshoppers, thrips, and whiteflies. Furthermore, we successfully obtained the full genome of SFSolV1 and observed a 100% infection rate in our laboratory WBPH populations. Moreover, our investigation into the infection and distribution characteristics of SFSolV1 demonstrated that this virus is more abundant in male adults compared to female adults. These findings highlight the need for further research to better understand the interactions between this virus and insects.

**Abstract:**

Agricultural insects play a crucial role in transmitting plant viruses and host a considerable number of insect-specific viruses (ISVs). Among these insects, the white-backed planthoppers (WBPH; *Sogatella furcifera*, Hemiptera: Delphacidae) are noteworthy rice pests and are responsible for disseminating the southern rice black-streaked dwarf virus (SRBSDV), a significant rice virus. In this study, we analyzed WBPH transcriptome data from public sources and identified three novel viruses. These newly discovered viruses belong to the plant-associated viral family *Solemoviridae* and were tentatively named Sogatella furcifera solemo-like virus 1-3 (SFSolV1-3). Among them, SFSolV1 exhibited a prevalent existence in different laboratory populations, and its complete genome sequence was obtained using rapid amplification of cDNA ends (RACE) approaches. To investigate the antiviral RNA interference (RNAi) response in WBPH, we conducted an analysis of virus-derived small interfering RNAs (vsiRNAs). The vsiRNAs of SFSolV1 and -2 exhibited typical patterns associated with the host’s siRNA-mediated antiviral immunity, with a preference for 21- and 22-nt vsiRNAs derived equally from both the sense and antisense genomic strands. Furthermore, we examined SFSolV1 infection and distribution in WBPH, revealing a significantly higher viral load of SFSolV1 in nymphs’ hemolymph compared to other tissues. Additionally, in adult insects, SFSolV1 exhibited higher abundance in male adults than in female adults.

## 1. Introduction

Insects, as vectors for numerous significant viruses, are considered to play a crucial role in the occurrence and prevalence of pathogenic viruses in humans, animals, and plants [1,2,3]. Recent advancements in next-generation sequencing technology and bioinformatics analysis tools have brought about a new era of virus discovery. Consequently, a considerable number of viruses lacking obvious pathogenicity have been identified in insects, many of which are known as insect-specific viruses (ISVs) [2]. ISVs are abundant, and diverse and engage in complex interactions with host insects, enhancing our understanding of the insect virome and insect–virus coevolution [4,5,6]. Numerous insect-borne pathogenic viruses together with ISVs can infect and proliferate within insects, indicating a close phylogenetic relationship between them. For instance, the majority of mosquito-borne viral pathogens infesting animals and humans belong to the same order/family as mosquito-specific viruses, such as *Flaviviridae*, *Togaviridae*, and *Phenuiviridae* [5,7,8]. Therefore, ISVs are likely to act as potential sources of arboviruses.

In recent years, researchers have gradually discovered viruses in phytophagous insects such as whiteflies, grasshoppers, leafhoppers, thrips, and planthoppers [3,9,10,11,12]. These viruses primarily belong to several taxa, including *Baculoviridae*, *Parvoviridae*, *Flaviviridae*, *Ascoviridae*, *Togaviridae*, *Bunyavirales* (formerly known as *Bunyaviridae*), and *Rhabdoviridae* [3]. The white-backed planthopper (WBPH), *Sogatella furcifera* (Hemiptera: Delphacidae), is one of the significant pests in rice cultivation. Several ISVs have been identified and reported in WBPH, primarily classified into the insect-associated virus families *Iflaviridae* and *Dicistroviridae* [13], the vertebrate-related virus family *Hepeviridae* [14], and the fungi/protozoa-related virus family *Totiviridae* [15]. WBPH transmits the southern rice black-streaked dwarf virus (SRBSDV) in a persistent, propagative manner [16]. With the long-distance migration of insects, this virus has become widely distributed in southern China, northern Vietnam, Japan, and Thailand, making it one of the most serious rice viruses [17]. However, no relevant research exists exploring the evolutionary relationship between ISVs in WBPH and plant viruses.

In this study, we conducted a transcriptome analysis using public databases from NCBI and identified three viruses belonging to the *Solemoviridae* family in WBPH, named Sogatella furcifera solemo-like virus 1-3 (SFSolV1-3). *Solemoviridae* is a plant-associated viral family, and its members have the ability to infect various plants, including dicots and monocots [18]. Some solemoviruses, such as rice yellow mottle virus, potato leaf roll virus, and sugarcane yellow leaf virus, are pathogenic and are causal agents of severe diseases of economically important plants. Additionally, certain members of this family are transmitted by insects [19,20,21,22]. The prevalence of the three solemo-like viruses discovered in WBPH was investigated in various laboratory populations by examining RNA-seq databases available at GenBank. Among these viruses, SFSolV1 exhibited a 100% infection rate in our laboratory populations. The complete genome sequences of SFSolV1 were obtained through rapid amplification of cDNA ends (RACE). Additionally, the analysis of virus-derived small interfering RNA (vsiRNA) revealed that both SFSolV1 and SFSolV2 can activate the RNA interference (RNAi) antiviral immune pathway in the insect host. The infection and distribution of SFSolV1 in WBPH were also characterized.

## 2. Materials and Methods

### 2.1. Data Generation and Virus Discovery

A total of 47 publicly available raw transcriptome datasets of *Sogatella furcifera* retrieved from the NCBI Sequence Read Archive (SRA) repository (Appendix A) were trimmed of adaptor sequences for quality using FastQC and Trimmomati. The clean reads were de novo reassembled/assembled using the two assembler software packages Trinity (--seqType fq --max_memory 20G --CPU 36) and metaSPAdes (--cov-cutoff auto --rna) with default parameters [23,24]. For the discovery of potential viruses, the assembled contigs were aligned with the NCBI nonredundant (NCBI-NR) databases using diamond blastx, with a cutoff E-value < 1 × 10^−20^ [25]. To eliminate false positive matches, viral homology contigs with sequence lengths of at least 1000 bp were chosen for further BLAST analysis against the whole NCBI nucleotide (NT) and non-redundant (NR) protein databases. Moreover, the viral homology contigs need to meet the criterion of containing complete open reading frames (ORF) of predicted viral RNA-dependent RNA polymerase (RdRP). The viral homology contigs needed to be confirmed by both of the assemblers, Trinity and metaSPAdes.

To investigate the distribution of the three solemo-like viruses, the adaptor- and quality-trimmed reads from each of the public transcriptome dataset deposited by seven universities or institutions were mapped back to viral contig using Bowtie2 and Samtools [26]. The coverage of viruses > 10 was confirmed as virus-positive.

### 2.2. Virus Detection

As the three solemo-like viruses were identified from public transcriptome data, we conducted detections to confirm the presence of them in our lab WBPH population. We took 15 adult and 15 nymph insects from a lab population and performed total RNA extraction and RT-PCR. Sanger sequencing was followed with the primers specified in Appendix A.

### 2.3. Insect Samples

The lab-reared population of WBPH was kept on healthy rice seedlings in climate-controlled rooms in Ningbo University at 25 ± 1 °C with 60 ± 5% relative humidity and 16 h light/8 h dark.

### 2.4. Virus Genome Determination

The rapid amplification of cDNA ends (RACE) with SMARTer^®^ RACE 5′/3′ kit (Takara, Dalian, China) was conducted to get the full genome of SFSolV1 according to the manufacturers’ instructions. The RACE PCR was carried out with high-fidelity polymerase (Phanta Max Super-Fidelity DNA Polymerase, Vazyme Biotech) under the following reaction conditions: 95 °C, 3 min; 95 °C, 30 s, 72 °C, 30 s, 72 °C, 30 s, 10 cycles; 95 °C, 30 s, 60 °C, 30 s, 72 °C, 30 s, 10 cycles; 72 °C, 10 min; 16 °C, hold. The PCR products were cloned into the pClone007 vector and Sanger sequenced. The primers used for RT-PCR and RACE are listed in Appendix A.

### 2.5. Phylogenetic Analysis

The ORFs of viral genomes were predicted using NCBI ORFfinder (https://www.ncbi.nlm.nih.gov/orffinder, accessed on 22 February 2024). For each ORF of the newly discovered solemo-like viruses, the protein functions were annotated with InterPro [27]. For phylogenetic analysis, the conserved RdRP of the three novel viruses combined with RdRP protein sequences from reference viruses were used. For reference viruses, we selected the representatives from *Solemoviridae*, *Tombusviridae*, *Dicistroviridae*, and *Iflaviridae* in order to clarify the taxonomic status of these three viruses in this study. The RdRP sequences were aligned with MAFFT, and ambiguously aligned regions were then trimmed by Gblock [28,29]. ModelTest-NG was used to determine the best-fit model of amino acid substitution. RAxML-NG was used to create maximum likelihood (ML) trees using 1000 bootstrap replications [30]. Details of all the reference sequences used in phylogenetic analysis are listed in Appendix A.

### 2.6. RNA Extraction and Sequencing

Total RNA extraction from lab-reared WBPH was performed using TRIzol (Invritrogen, Carlsbad, CA, USA). The extracted RNA was sent to Novogene, Tianjin, China, for transcriptomic and small RNA (sRNA) sequencing. A paired-end (150 bp) library was constructed and sequenced on the Illumina HiSeq4000 platform (Illumina, CA, USA) for transcriptomic sequencing. For sRNA sequencing, the cDNA library was prepared using TruSeq Small RNA Sample Preparation Kit (Illumina, CA, USA), and sRNA sequencing was performed on an Illumina HiSeq 2500. The output raw sRNA reads were treated to remove the adaptor, low quality, and junk sequences using the CUtadapt tool as described previously [31].

### 2.7. Small RNA Analysis

To identify the virus-derived siRNAs (vsiRNAs), the 16–32 nt long quality-trimmed sRNA reads were extracted using the FASTX-Toolkit (http://hannonlab.cshl.edu/fastx_toolkit/, accessed on 27 February 2024) and then mapped back to the identified viruses in this study using Bowtie with zero mismatches as described previously [31]. The analysis scripts used in this article refers to the virus analysis pipeline on github (https://github.com/Gyoungwe/novel_virus_analysis, accessed on 28 February 2024).

### 2.8. The Distribution of SFSolV1 in Different Tissue Types of WBPH

To clarify the tissue-distribution pattern of SFSolV1 in WBPH, tissue samples from the 4^th^ to 5^th^ nymphs (including head, gut, fat body, hemolymph, and residues) and tissue samples from adult insects at 3–4 days post eclosion (including head, gut, fat body, hemolymph, residues, and female/male reproductive) were dissected and collected, and total RNA was extracted. For RT-qPCR assays, about 8–12 isolated tissue pieces were mixed together as one sample for RNA extraction and RT-qPCR, and a total of 4 replicates for each tissue type were subjected to RT-qPCR testing. Then, RT-qPCR was performed using virus-specific primers. The reaction protocol included an initial denaturation at 95 °C for 5 min, followed by 40 amplification cycles at 95 °C for 15 s, 60 °C for 15 s, and 72 °C for 20 s. The reaction concluded with a final sequence at 60 °C for 60 s and 95 °C for 15 s. Threshold cycle (Ct) values were normalized to the tubulin gene (GenBank accession number: KP735521.1) using the comparative 2^−ΔΔCt^ methods [32]. The primers of SFSolV1 used for in this experiment are listed in Appendix A.

### 2.9. Viral Detection of SFSolV1 in Different Potential Hosts

In order to determine the potential host range of SFSolV1, the infectivity towards two closely related planthopper species, namely the brown planthopper (BPH) *Nilaparvata lugens* and the small brown planthopper (SBPH) *Laodelphax striatellus*, was evaluated, as described previously [11,33]. Ten 3^rd^-instar nymphs were selected from each planthopper species and placed on five rice seedlings for a period of 96 h. Afterward, the planthoppers were gathered to detect the presence of SFSolV1 using RT-PCR. The presence of SFSolV1 in rice seedlings was also detected, as described previously [34]. After the rice seedlings had been fed on by insects for 96 h, the rice seedlings were gathered as soon as the planthoppers were removed for SFSolV1 detection using RT-PCR. To eliminate any interference from viruses in the honeydew, each rice seedling was thoroughly cleaned with a sodium hypochlorite solution. This experiment was repeated three times. Since there was no SFSolV1-nagative WBPH, the healthy non-feeding rice seedlings were also detected as controls.

### 2.10. Statistical Analyses

Statistical analyses were conducted using Graphpad InStat 3. Student’s *t*-test or ANOVA with Tukey LSD tests were used to determine the level of significance in the relative levels of the SFSolV1.

## 3. Results

### 3.1. Discovery of Three Novel Solemo-like Viruses in WBPH

To identify viruses in WBPH, we collected a total of 47 transcriptome datasets from the NCBI SRA repository, which were submitted by seven universities or institutions in China (Appendix A). Following quality trimming and assembling the transcriptomes, we aligned the contigs with the NCBI-NT and -NR databases using rigorous criteria to identify novel viruses. Our analysis unveiled three previously uncharacterized viruses that contained a complete open reading frame (ORF) of a predicted RNA-dependent RNA polymerase (RdRP). These viruses were found to be related to members of the *Solemoviridae* family. Accordingly, we provisionally named them “Sogatella furcifera solemo-like virus 1” (SFSolV1), “Sogatella furcifera solemo-like virus 2” (SFSolV2), and “Sogatella furcifera solemo-like virus 3” (SFSolV3). Protein BLAST search against the NCBI database revealed that SFSolV1 had highest similarity to Atratosobemo-like virus 4 (QHA33876.1) with 52.5% of the sequence identities, SFSolV2 exhibited the highest similarity to Amygdalus persica sobemo-like virus (QKI29238.1) with 45% of the identity, and SFSolV3 was the most closely related to Soybean thrips sobemo-like virus 9 (QQ081419.1) with 38.7% sequence identity. The presence of these three viruses was detected in each dataset, with SFSolV1 being present in all seven lab populations, while SFSolV2 and SFSolV3 were only found in a subset of these lab populations (Figure 1A).

### 3.2. Genome Organization of the Novel Viruses

We employed RT-PCR to detect the presence of the three viruses in our lab populations. The results showed that the infection rate of SFSolV1 was 100% while SFSolV2 exhibited an infection rate of only 2.5%, and no infections were observed with SFSolV3 (Figure 1B). Consequently, we conducted RACE using the SMARTer RACE 5′/3′ Kit to obtain the full genome sequence of SFSolV1. The complete genome of SFSolV1 consisted of a polycistronic, positive-sense RNA molecule of 2792 nt, without a 3′-poly(A) tail (accession number: NMDCN000370F). It comprised two predicted open reading frames (ORFs) analyzed using NCBI ORFfinder. ORF1 (nt 39–1485) contained a peptidase domain, while ORF2 (nt 1457–2696) contained an RdRP (Figure 1C). Similar predictions were made for viral ORFs and protein functions in SFSolV2 (accession number: NMDCN000370G) and SFSolV3 (accession number: NMDCN000370H). Both SFSolV2 and SFSolV3 consisted of two ORFs, where ORF1 contained a peptidase domain and ORF2 contained an RdRP domain (Figure 1C). The RdRP nucleotide sequences of the three viruses revealed 26.1–27.2% identity (Figure 1C). These findings indicated that the genome organization of the three solemo-like viruses identified in this study was similar. However, no CP proteins were found, and their genome organizations differed significantly from those of known plant viruses in the *Solemoviridae* family [18]. Thus, further investigation is required to determine the evolutionary relationship of these three solemo-like viruses with plant viruses in the *Solemoviridae* family.

### 3.3. Phylogenetic Analysis of Novel Viruses

To further evaluate the taxonomical status of these three viruses, we conducted a phylogenetic analysis using the conserved RdRP domains of SFSolV1, SFSolV2, and SFSolV3, as well as the RdRP from other related sequences and representatives from *Solemoviridae*, *Tombusviridae*, *Dicistroviridae*, and *Iflaviridae*. The amino acid sequences of RdRPs were aligned, and a maximum likelihood (ML) tree was constructed. The phylogenetic analysis revealed that the three solemo-like viruses were grouped in an unclassified clade within the *Solemoviridae* family (Figure 2). RdRP is a crucial gene involved in virus replication and is considered the most conserved gene among all RNA viruses, making the phylogenetic tree of the RdRPs reliable [35,36]. Moreover, since certain plant viruses within this family are also transmitted by insects [18,19,20,37], it is highly likely that many other undiscovered members of this family also infect insects [36].

### 3.4. Profiles of Virus-Derived Small Interfering RNAs for Sogatella furcifera Solemo-like Viruses in WBPH

The RNA interference (RNAi) pathway mediated by small interfering RNA (siRNA), plays a crucial role as an innate antiviral defense mechanism in insects, triggered upon virus invasion and propagation [38]. In this study, we conducted sRNA sequencing on laboratory-reared WBPH and characterized the virus-derived siRNAs (vsiRNAs) of SFSolV1 and SFSolV2. These vsiRNAs displayed a typical size distribution and polarity, with a predominant peak at lengths of 21 and 22 nucleotides (nt) (Figure 3A,C). Moreover, the vsiRNAs derived from both the sense and antisense strands of the viral genomic RNA were nearly equivalent in proportion. In the two viral genomes, the vsiRNAs were symmetrically distributed, with some asymmetrical hotspots potentially serving as specific targets for Dicer (Figure 3B,D). These typical vsiRNA patterns indicate the active involvement of the host’s antiviral RNAi pathway in response to viral infection.

### 3.5. Tissue Distribution in WBPH and Host Range Detection of SFSolV1

In the previous detection (Figure 1B), SFSolV1 exhibited a 100% infection rate in the lab population of WBPH. Therefore, an investigation was undertaken to examine the infestation of SFSolV1 in various tissues of nymph and adult WBPH insects. Using RT-qPCR, we assessed the relative abundance of the virus in different nymph tissues including the head, gut, fat body, hemolymph, and residues. Our findings indicate the presence of SFSolV1 in various nymph tissues, with the hemolymph showing the highest abundance (Figure 4A). Additionally, we examined the abundance of SFSolV1 in tissues of adult insects, including the head, gut, fat body, hemolymph, and female/male reproductive system. The results revealed that SFSolV1 in adult insects exhibited significantly higher viral abundance in the gut, fat body, hemolymph, and male reproductive system when compared to female insects. No significant differences were observed in the head between male and female adults (Figure 4B). Consequently, the abundance of SFSolV1 is expected to be higher in males than in females at the whole-insect infection level.

Considering the close relationship between WBPH and two other planthopper species, the brown planthopper (BPH), *Nilaparvata lugens*, and the small brown planthopper (SBPH), *Laodelphax striatellus*, we investigated the presence of SFSolV1 in these two planthopper species. The third-instar nymphs of BPH and SBPH were co-fed with SFSolV1-positive WBPH on healthy rice seedlings for 96 h, as described previously [33]. Afterward, the nymphs were collected, and virus detection was carried out using RT-PCR. The results revealed the absence of SFSolV1 in both BPH and SBPH individuals (Figure 4C, Appendix A). Additionally, no SFSolV1 was detected in the rice seedlings that were used for feeding (Figure 4C, Appendix A). These findings indicate the SFSolV1 is incapable of infecting the other two species of planthoppers or rice plants.

## 4. Discussion

This study identified three novel insect-specific viruses (ISVs) belonging to the plant-associated viral family *Solemoviridae* in *Sogatella furcifera*. Among them, SFSolV1 was prevalent in various laboratory samples (Figure 1A). Additionally, vsiRNA analyses confirmed the specific infestation of SFSolV1 and -2 in WBPH, as well as their ability to trigger an RNAi antiviral immune response in the insects (Figure 3). Although the members of the *Solemoviridae* family are typically characterized as plant viruses according to the ICTV report, numerous novel ones have recently been identified in invertebrates through metagenomic approaches [39]. With the revelation of virus diversity in insects, including strips, mosquitoes, and wasps, many viruses were discovered and characterized as solemo-like viruses depending on the phylogenetic tree of the RdRPs [12,40,41]. Phylogenetic analysis in this study revealed that the three novel solemo-like viruses belong to unclassified clades within the *Solemoviridae* family, indicating the potential existence of novel genera. Plant viruses are proposed to have originated from invertebrate viruses [36], and many of them are transmitted by insect vectors. Therefore, we hypothesize that there may be numerous undiscovered solemo-like viruses related to insects or other arthropods.

Recent advancements in insect virome research have also revealed a close evolutionary relationship between insect-transmitted pathogenic viruses and ISVs [1,5]. For example, mosquitoes harboring ISVs of the same classification as mosquito-borne viruses have been identified, such as viruses from *Flaviviridae*, *Phenuiviridae*, and *Bunyavirales* [42,43,44,45]. Similarly, the rice planthoppers *Laodelphax striatellus* and *Nilaparvata lugens* transmit rice reoviruses and also harbor insect-specific reoviruses [34,46]. Within the *Solemoviridae* family, several viruses significantly infest rice and cause severe disease, such as the rice yellow mottle virus (RSMV) and a novel virus called rice tiller inhibition virus (RTIV) [21,37]. The planthopper WBPH feeds on rice and transmits the important rice viruses SRBSDV. The discovery of solemo-like viruses in WBPH suggests that planthoppers may also serve as vectors for solemoviruses that infest plants.

Insects harbor numerous viruses, including ISVs, insect-borne viruses, and viruses derived from the living environment [47]. Here, the vsiRNA analysis demonstrated that the two viruses SFSolV1 and SFSolV2 were hosted by WBPH. Insects have evolved both systemic and tissue-specific antiviral mechanisms in response to viral infection and replication [48]. Conversely, in order to adapt to the host environment, viruses depend on either host factors or their own factors present within their harboring tissues of insects to evade triggering a defensive immune response, thus maintaining a state of relative homeostasis [44,48]. As a result, the tissue-expression pattern of viruses in host insects reflects the interaction between the virus and the host. In this study, distinct tissue preferences for SFSolV1 are observed between nymph and adult insects (Figure 4), which may be associated with insect development. During the investigation of infestation by white-backed planthoppers (WBPHs), we observed a 100% infestation rate in our laboratory population (Figure 1B), suggesting the likelihood of vertical transmission through transovarial means, whereby the virus is transmitted from the mother to her offspring [49,50]. In our study, while SFSolV1 was relatively scarce in the female reproductive system, it exhibited abundance in the male reproductive system. Considering some exceptional cases, such as a symbiotic virus RdFV in leafhoppers, sigma virus in Drosophila melanogaster, and Diaphorina citri reovirus (DcRV) in psyllids [51,52,53], viruses can also be transmitted paternally. Therefore, further investigation is necessary to determine the exact mechanism of vertical transmission for this virus.

ISVs are believed to only replicate in host insects and exhibit persistent infestation. Both vertical and horizontal transmission of ISVs are thought to exist in host insect populations [1,2]. Symbiotic ISVs display intricate interactions with their host insects, despite not causing evident pathogenic symptoms [1,4]. For instance, symbiotic partiti-like viruses identified in African armyworms have been shown to reduce the growth rate and reproductive capability of the host insect while concurrently increasing their resistance to a nucleopolyhedrovirus (NPV) [54]. Another example involves a -ssRNA virus found in parasitoid wasps, which decreases the proportion of female offspring in host insects [55]. Our findings unveil a significant difference in the abundance of SFSolV1 between male and female adult WBPHs. SFSolV1 was found to be more prevalent in male adults compared to their female counterparts. These outcomes warrant further investigation into the potential influence of SFSolV1 on the reproductive capabilities of WBPH populations, as well as its potential gender-specific effects. This research will contribute to a better comprehension of the dynamics between ISVs and their hosts, thus facilitating the development of effective pest-management strategies in agriculture.

## Figures and Tables

**Figure 1 insects-15-00394-f001:**
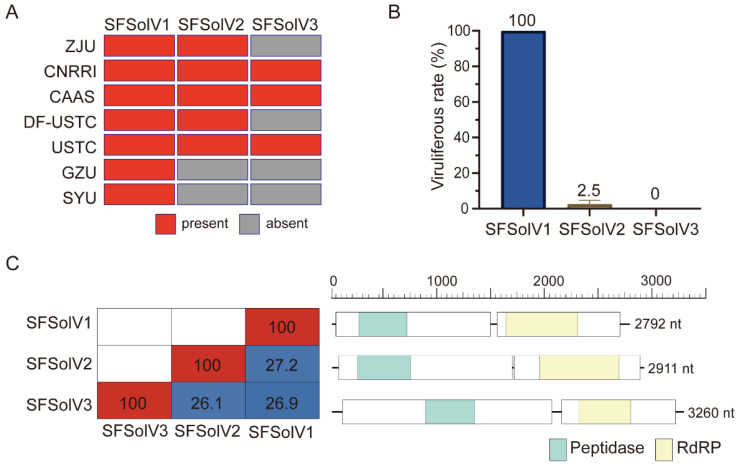
The presence, detection, and genomic organizations of three novel solemo-like viruses identified in white-backed planthopper (WBPH). (**A**) The presence of each solemo-like virus across different lab populations. SFSolV1 was present in all seven lab samples based on the mapping analysis of transcript reads, SFSolV2 was present in five lab samples, and SFSolV3 was present in three lab samples. (**B**) The detection of three viruses in our lab-cultured populations. SFSolV1 exhibited an infection rate of 100%, the infection rate of SFSolV2 was only 2.5%, and the SFSolV3 was absent. (**C**) Average RdRP nucleotide identity among three solemo-like viruses and the viral genomic organizations. The pairwise distances between RdRP nucleotide sequences (left) among the three solemo-like viruses calculated using MegAlign from the DNASTAR Lasergene 12 software package. The genomic organizations of three viruses (right) all comprised two ORFs. ORF1 contained a peptidase domain, and ORF2 contained an RdRP domain.

**Figure 2 insects-15-00394-f002:**
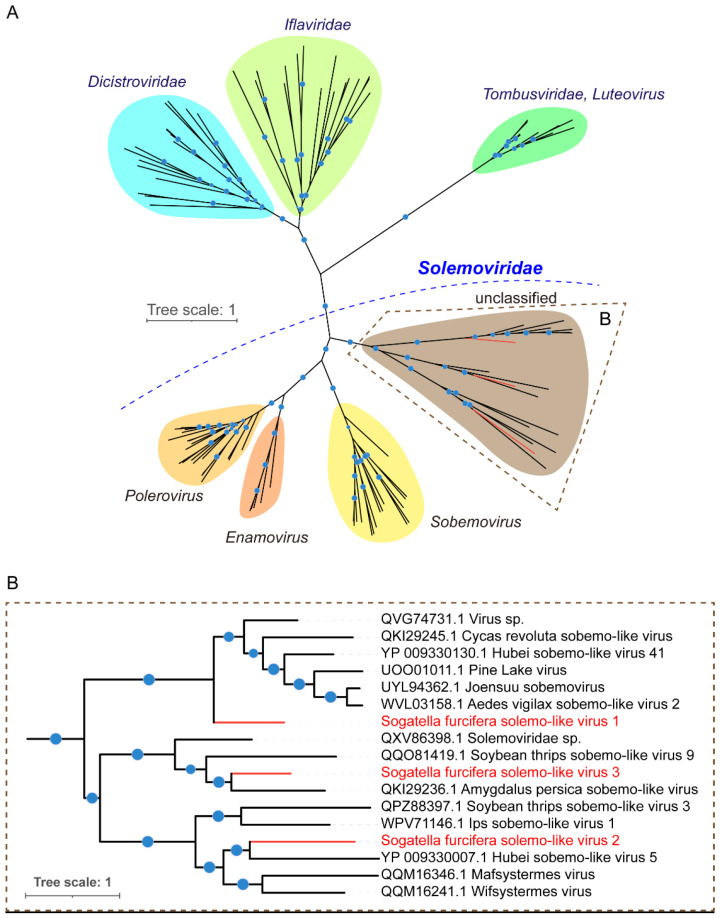
(**A**) Phylogenetic trees of the three solemo-like viruses identified in white-backed planthopper (WBPH). (**B**) The maximum-likelihood phylogenetic tree was created based on conserved viral RdRP domains. The viruses SFSolV1-3 are shown in red front. Nodes with bootstrap values >70% are marked with blue circles, and the larger circles indicate higher bootstrap values.

**Figure 3 insects-15-00394-f003:**
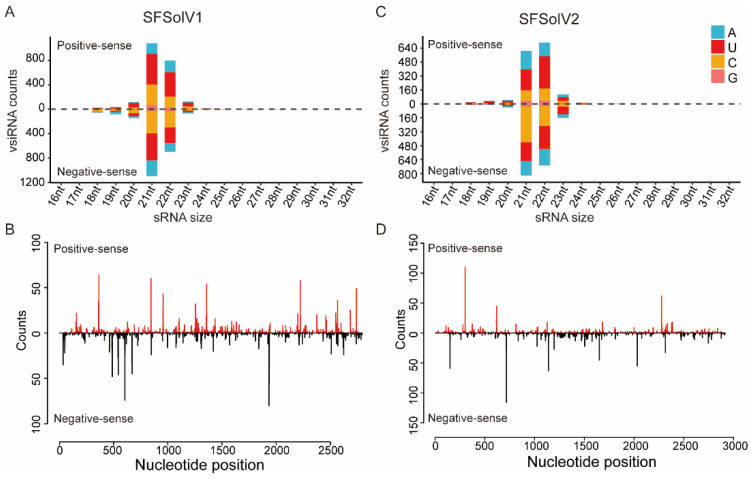
Profiles of Sogatella furcifera solemo-like virus derived sRNAs. The sized distribution of SFSolV1- (**A**) and SFSolV2- (**C**) derived sRNAs, and the distribution of vsiRNAs corresponding to the viral genome of SFSolV1 (**B**) and SFSolV2 (**D**) were analyzed.

**Figure 4 insects-15-00394-f004:**
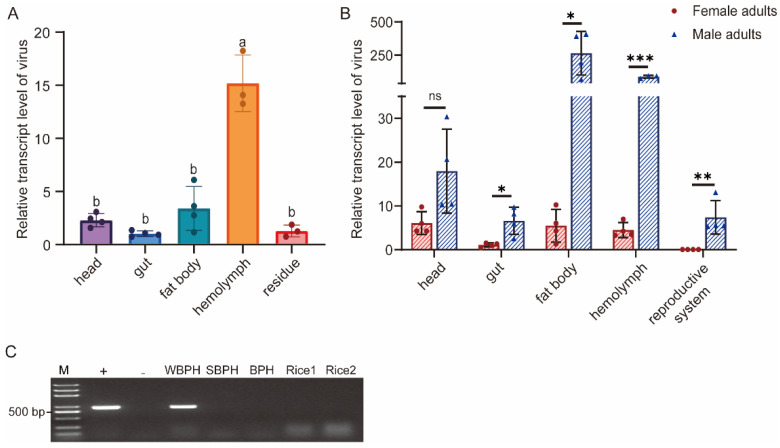
The distribution and abundance of SFSolV1 in different tissues of nymph and adult white-backed planthopper (WBPH). (**A**) Relative quantification of SFSolV1 in different tissues of nymph insects, including head, gut, fat body, hemolymph, and residue. Data are represented as means ± SEM of four biological replicates per sample. Different letters on the error bar indicate significant differences (*p* < 0.05, one-way ANOVA). (**B**) Relative quantification analysis and comparison of SFSolV1 between female and male adults in different tissues, including head, gut, fat body, hemolymph, and reproductive system. (**C**) Host range detection of SFSolV1. Virus detection in BPH and SBPH, which were fed together with SFSolV1-positive WBPH on rice seedlings. This electrophoretogram is representative as three repetitions. M, DNA marker; +, positive control; -, negative control; Rice 1, feeding rice seedlings; Rice 2, non-feeding rice seedlings. * *p* < 0.05; ** *p* < 0.01; *** *p* < 0.001; ns, not significant.

## Data Availability

Sequences of all identified novel viruses from this study have been deposited in Nature Microbiology Data Center (NMDC) with accession numbers NMDCN000370F, NMDCN000370G, and NMDCN000370H. The raw reads of the sRNA-seq generated in this study were deposited in NMDC with accession number NMDC40052737. The authors declare that all data supporting the findings of this study are available in the manuscript, and its Appendix A files are available from the corresponding authors upon request.

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
