# Peer review of "Identification and Characterization of Three Novel Solemo-like Viruses in the White-Backed Planthopper, *Sogatella furcifera"

_insects, 2024, doi:10.3390/insects15060394_

Round 1

Reviewer 1 Report (Previous Reviewer 1)

Comments and Suggestions for Authors

Line 249:It is more appropriate to change the title to “Tissue distribution in WBPH and host range detection of SFSolV1”

Author Response

Many thanks for your positive evaluation of our work and valuable
suggestions and we have followed.

Reviewer 2 Report (New Reviewer)

Comments and Suggestions for Authors

In this study, Yuan et al. reported 3 new solemo-like viruses from mining earlier published transcriptome data. Overall, the authors need to explain the significance of their findings beyond simply reporting their experimental results. Several result sections lack proper interpretations and sections of the methodology need additional detail, citations etc. In the current state, the writing quality is also low and overall tone of the manuscript seems rushed. Therefore, at this stage I cannot recommend this manuscript for acceptance. Below are my specific comments:

Lines 94-104: The transcriptome analyses programs used by the authors needs proper citations. What were the “default” parameters used for de novo assembly?

The analyses conducted on lab populations of WBPH need to be in a separate section. It does not make sense to combine that with section 2.1. and 2.2. in methodology.

How is section 2.2. different from 2.1.? Virus “detection” vs virus “discovery” in the same transcriptome data set and lab populations of WBPH?

The authors need to clearly distinguish in their writing that the findings from earlier transcriptome data is independent of their findings from lab populations of WBPH – this is not clear in the current version of methodology.

Section 2.4. mentions genome sequence completion only for SFSolV1 and not the other viruses. However, table S2 lists primers for genome amplification of the other viruses as well. Authors need to indicate primer coordinates w.r.t. the respective viral genomes. Use of the RACE kit makes sense for determining the accurate 5’- and 3’- ends of the genome but generating the complete genome sequence for these 2.7-3.2 kb viruses would require RT-PCRs with high-fidelity polymerases etc. There is no mention of what RT-PCR/RT-qPCR parameters were used in this study. Section 2.4. needs more detail overall and the phylogenetic analyses should be a separate section.

Line 148-149: “The output vsiRNAs were further calculated with custom perl script and the Linux bash scripts.” What does this mean?

Section 2.7. The title should indicate that this was performed in different types of WBPH tissue. Needs more clarity in terms of number of replicates vs. number of insects pooled into one sample per tissue type. What RT-qPCR conditions were used?

Section 2.8. what was the inoculum source of SFSolV1 for the nymphs? Did the authors simply rear healthy nymphs on healthy rice seedlings? Define “replicate” for this section.

Section 3.1. What is the level of identity of these viruses to members of Solemoviridae? How do they compare to other solemo-like viruses? These are important points of reference for interpreting the sequence data.

Section 3.2. Were these viruses spontaneously detected in lab populations of WBPH with no apparent source of the inoculum? This needs explaining.

Figure 1C. What genome region does the ID matrix represent? There is no explanation.

In Figure 2, I would like to see only the branches with bootstrap support >70%. In the current form, I am not convinced that these new viruses are closely related to Solemoviridae. In 2B, the viruses indicated in black color seem to be distantly related to Solemoviridae in general, except QXV86398.1 (which is still, unclassified). This phylogram would benefit from inclusion of other solemo-like viruses.

Line 229-231: “it is highly likely that many other undiscovered members of this family also infect insects”. Citations needed.

Section 3.4. RNAi as a defense mechanism against viruses has been well-studied. Therefore, it is not surprising that these solemo-like viruses trigger RNAi in the host WBPH. However, is there any significance of these findings beyond the fact that RNAi is occurring in the host insect? Authors should highlight why this piece of information is valuable.

In Figure 3, for B and D, reads mapping to negative sense RNA should be clearly indicated; showing counts under a negative scale does not make sense.

Section 3.5. Where did the SFSolV1 come from in the lab population of WBPH? Was it previously undetected? Is this a common phenomenon seen for ISVs that they are present in lab-reared experimental insect populations without an apparent source of the infection evident? This needs explaining, citations, etc. Authors mentioned insect-symbiotic viruses in the discussion – has this been seen in the case of other solemo-like viruses?

Lines 270-272: This statement is not supported by enough experimental evidence. How long did the researchers wait before testing the rice seedlings for SFSolV1? A proper time course experiment needs to be designed, that includes rice along with model plant species, to say that this virus is "incapable of infecting rice". Re: the two other planthopper species, positive controls need to be included to ensure that the experimental set up itself does not hamper virus uptake by the insects. This control was missing.

Before declaring these solemo-like viruses as part of Solemoviridae, authors need to present enough evidence that is in line with ICTV criteria for virus classification (nucleotide and amino acid identity levels of relevant ORFs/proteins being within the cut-off limits for family/genus-level demarcations). Here they only present a phylogram which is not enough. They should compare their findings with other reports of solemo-like viruses to highlight significance of their study.

Comments on the Quality of English Language

There are typos and grammatical errors throughout the main text that require significant correcting and improvement. There are statements in the results sections which are hard to understand due to issues in sentence structure.

Round 2

Reviewer 2 Report (New Reviewer)

Comments and Suggestions for Authors

The overall quality of the manuscript after revision by Yuan et al. has improved significantly from round-1. I believe it can be recommended for acceptance after clarifications have been provided by authors to my comments below:

Formatting of the PCR parameters needs improvement throughout the manuscript. Authors should refer other publications and re-write.

Lines 167-169: should be re-written as- “For RT-qPCR assays, about 8-12 isolated tissue pieces were mixed together as one sample for 167 RNA extraction and RT-qPCR, a total of 4 replicates for each tissue type were subjected to RT-qPCR testing”.

Line 172: Citation needed for the 2-ΔΔCt method.

Section 2.9. It is still not clear where the inoculum source of SFSolV1 for the nymphs was in this assay. The paper cited in this section describes a similar experiment where the candidate insect nymphs were fed on an artificial diet containing the novel virus and then transferred to rice seedlings. However, the authors state that in this study they performed a “co-feeding” assay which is different from the cited paper. Clarification is needed here with an appropriate paper cited, explanation of the assay method, and the experimental controls especially. In page 5 of their revised cover letter, authors state “The rice seedlings were detected as soon as the planthoppers were moved. Considering that the SFSolV1-positive WBPH has been feeding on rice seedlings for 96 h and the detection of SFSolV1 on rice seedling is still negative”. Is 96h sufficient incubation period for detection these viruses? In my experience, one needs to monitor receptor plants/seedlings for up to several weeks/months after completion of the insect transmission assay. Kindly correct me if my understanding of the methodology used in this study is incorrect.

Lines 221-222: Authors should say “genome organization” instead of “structure” when they talk about similarity between the 3 viruses. “Structure” can be inferred to mean virion particle structure. Did they investigate this via EM or similar techniques?

Lines 222-224: If no CP-coding ORFs were detected in the novel viruses’ genome sequences, what is the meaning of “their structures differed significantly from those of 223 known plant viruses in the Solemoviridae family”.

Figure 2. Have the authors tried phylogenetic analyses with the inclusion of other reported solemo-like viruses? An example is here: Litov, A.G., Belova, O.A., Kholodilov, I.S., Gadzhikurbanov, M.N., Gmyl, L.V., Oorzhak, N.D., Saryglar, A.A., Ishmukhametov, A.A. and Karganova, G.G., 2021. Possible arbovirus found in virome of Melophagus ovinus. Viruses13(12), p.2375. The above paper reports solemo-like viruses from a mammalian ectoparasite and describes some phylogenetic analyses. Since this study is doing the same, the phylogenetic analyses would have benefited from the inclusion of previously reported solemo-like viruses.

In their revised cover letter (page 6), the authors refer to the ICTV Lispiviridae report in response to my comment from review round-1. I did not understand how this answered my question. I understand that phylogenetic analyses using the RdRp protein sequence is significant for explaining the diversity of RNA viruses with the authors providing citations in line 326 having examples of sobemo-like viruses. My original comment was asking for evidence supporting these solemo-like viruses to be classified as a new genus of Solemoviridae, and my suggestion was to compare the viruses from this study to other solemo-like viruses reported in other studies. If the authors believe that analysis is not necessary, then that should be explained in their results/discussion.

Lines 296-298: The cited paper does not describe a co-feeding assay. Please correct this section and the respective section in methodology.

The discussion has been significantly improved with the new written sections and citations.

There are still typos in the manuscript – Line 19: “viRNA” should be vsiRNA; Line 234: “basent” should be absent. These are some examples and there are more in the main text.

Comments on the Quality of English Language

Quality of writing has not improved since round-1 of review. There are still issues with sentence structure and typos in the main text. The authors should consider using a language editing program or a professional service.

Author Response

This manuscript is a resubmission of an earlier submission. The following is a list of the peer review reports and author responses from that submission.

Round 1

Reviewer 1 Report

Comments and Suggestions for Authors

The paper reports three novel solemo-like viruses in the insect. It provides important clues for the study of insect-specific viruses. However, there are also some questions below: 

1. Has the RNA sequencing data used in the article been uploaded to public databasesIf submitted please provide the accession number.

 2. In this article, the authors reported three novel viruses in the insect, which assigned to plant-reported viral family. In the discussion, line278-280, It's also thought to infect plants, so how do author know it's an insect virus, but not a plant virus.

 3. Whether these viruses can be detected in rice or other rice planthoppers, such as Nilaparvata lugens and Laodelphax striatellus.

 4. What is the difference between these three viruses and what is the basis for dividing them into three species?

 5 Do these insect viruses have an impact on the developmental duration, fertility or male to female ratio of populations in different laboratories, as they contain different viruses? 

Author Response

The paper reports three novel solemo-like viruses in the insect. It provides important clues for the study of insect-specific viruses. However, there are also some questions below: 

  1. Has the RNA sequencing data used in the article been uploaded to public databases?If submitted please provide the accession number.

Response: Thank you for pointing out this. The RNA sequencing data has been uploaded to Nature Microbiology Data Center (NMDC) with accession number NMDC40052737, and we added this information in Lines 380-381.

  1. In this article, the authors reported three novel viruses in the insect, which assigned to plant-reported viral family. In the discussion, line278-280, It's also thought to infect plants, so how do author know it's an insect virus, but not a plant virus.

Response: Thank you for raising this issue. In the revised manuscript, we conducted host range detection of SFSolV1, one of the three solemo-viruses, because this virus exhibited a 100% infection rate in our lab population. Virus detection was performed using RT-PCR on rice seedlings fed on by WBPH, and no SFSolV1 was detected. This result indicate that SFSolV1 cannot infect rice plants. Related contents appeared in Lines 167-174, 287-290 and Figure 4C.

  1. Whether these viruses can be detected in rice or other rice planthoppers, such as Nilaparvata lugens and Laodelphax striatellus.

Response: Thank you. We conducted host range detection of SFSolV1, one of the three solemo-viruses, because this virus exhibited a 100% infection rate in our lab population. We investigated the presence of SFSolV1 in these two planthopper species. The 3rd instar nymphs of BPH and SBPH were co-fed with SFSolV1-positive BPH on rice seedlings for 96 hours. Afterward, the nymphs were collected, and virus detection was carried out using RT-PCR. The results revealed the absence of SFSolV1 in both BPH and SBPH individuals. Thus, SFSolV1 was indicated that it’s incapable of infecting the other two species of planthoppers. Related contents appeared in Lines167-174, 280-286 and Figure 4C.

  1. What is the difference between these three viruses and what is the basis for dividing them into three species?

Response: Thank you for your suggestion. In the revised manuscript, we compared the RdRp nucleotide sequences of these three viruses and they showed 26.1-27.2% identity. The related contents appeared in Figure 1C and Lines 208-209. 

 5 Do these insect viruses have an impact on the developmental duration, fertility or male to female ratio of populations in different laboratories, as they contain different viruses? 

Response: Thank you for your suggestion. In this manuscript, we focused on the identification and the genetic characterization of novel viruses. The functions of the viruses, such as impact on the developmental duration, fertility or male to female ratio of populations would be investigated further.

Reviewer 2 Report

Comments and Suggestions for Authors

Yuan et al have analyzed the transcriptome data sets of WBPH available with GenBank SRA repository and identified/characterized the genomes of three novel viruses and classified them as Solemo-like viruses. However, the viruses described in this study have a much reduced genome and completely different genome organization than that known of the viruses belonging to Solemoviridae. It is unclear how the authors decided that the novel virus genomes described in this study should be considered within Solemoviridae. The phylogenetic analysis of these three viruses with other genera within solemoviridae (fig2) using the RDRP gene is inappropriate as the there is no way for the readers to know why the authors chose these virus groups for the analysis. The RDRP of the viruses of solemoviridae are much larger and fusion proteins of the ORF 1 and 2 and ORF2b for sobemoviruses and I could not find details of identity of the RDRP of the identified viruses with them. thus, I think that at the present format, it is not appropriate to classify the described viruses as solemo-like viruses.

Other comments:

Line 80: replace 'can cause serious illness' with 'are causal agents of severe diseases of economically important plants'

Line 84: from RNA seq databases available at GenBank.

Line 85: Replace 'confirmed' with 'obtained'

Line 121: And Sanger sequenced.

Comments on the Quality of English Language

The article needs English editing as some indicated in the comments section.

Author Response

Yuan et al have analyzed the transcriptome data sets of WBPH available with GenBank SRA repository and identified/characterized the genomes of three novel viruses and classified them as Solemo-like viruses. However, the viruses described in this study have a much reduced genome and completely different genome organization than that known of the viruses belonging to Solemoviridae. It is unclear how the authors decided that the novel virus genomes described in this study should be considered within Solemoviridae. The phylogenetic analysis of these three viruses with other genera within solemoviridae (fig2) using the RDRP gene is inappropriate as the there is no way for the readers to know why the authors chose these virus groups for the analysis. The RDRP of the viruses of solemoviridae are much larger and fusion proteins of the ORF 1 and 2 and ORF2b for sobemoviruses and I could not find details of identity of the RDRP of the identified viruses with them. thus, I think that at the present format, it is not appropriate to classify the described viruses as solemo-like viruses.

Response: We appreciate the comments. In the revised manuscript, to further evaluate the taxonomical status of the three viruses in this study, we conducted the phylogenetic analyses using more reference viruses, including representatives from virus families Solemoviridae, Alvemaviridae, Tombusviridae, Dicistroviridae, and Iflaviridae. And the result showed that the three viruses exhibited a close relationship with members from Solemoviridae and were grouped in an unclassified clade. We acknowledge that the three viruses here are different from known plant-infected members of Solemoviridae, but as described in Lines 239-241, RdRP is recognized as the only gene that is conserved in all RNA viruses. Thus, the phylogenetic tree of the RdRPs is used as a scaffold to reconstruct the RNA virus evolution and develop the corresponding taxonomy. The related references were added. Therefore, the three viruses were described as solemo-like viruses. The related contents appeared in Lines 132-134 and 233-236.

Other comments:

Line 80: replace 'can cause serious illness' with 'are causal agents of severe diseases of economically important plants'

Response: Thank you, we revised it.

Line 84: from RNA seq databases available at GenBank.

Response: Thank you, we revised it.

Line 85: Replace 'confirmed' with 'obtained'

Response: We have replaced.

Line 121: And Sanger sequenced.

Response: We corrected it.

The article needs English editing as some indicated in the comments section.

Response: Thank you. We revised it.